# Fruit Anatomy of the Canarieae (Burseraceae)

**DOI:** 10.3390/plants11030253

**Published:** 2022-01-19

**Authors:** María C. Martínez-Habibe

**Affiliations:** Chemistry and Biology Department, Universidad del Norte, Barranquilla 08001, Colombia; mhabibe@uninorte.edu.co

**Keywords:** articulated plate, Burseraceae, Canarieae, fruit anatomy, resin, Sapindales, secretory canals

## Abstract

Fruits historically have been the key character for delimitation of tribes in the Burseraceae. However, fruit structure is incompletely known within the family, thus the importance of this character is unclear. This study of fruit anatomy in the traditional tribe Canarieae examines the distribution of the tissues that correspond to the exo-, meso-, and endocarp. The detailed arrangement and measurement of the tissues are reported here for the first time in all eight genera in the tribe. The evidence suggests that in all cases except *Pseudodacryodes*, the endocarp has at least one layer of parenchyma cells within which a sclereid layer is evident and, in some cases, an inner epidermis. All Canarieae fruits exhibit secretory canals, and some taxa have epidermal glands with resin-like contents. Evidence of carpellar sutures was found for all Canarieae, and in *Dacryodes*, *Haplolobus*, *Rosselia*, and *Santiria*, an articulated plate is present that corresponds to an abortive locule. The anatomical and morphological characters presented here are useful in delimiting genera within Canarieae.

## 1. Introduction

Infrafamiliar classification of Burseraceae historically has been defined primarily by fruit characters, especially the degree of connation of the pyrenes (a pyrene is a seed with bony endocarp) in the compound drupe. Engler’s two tribal classifications of Burseraceae [1,2] used the configuration of the pyrenes as the only diagnostic character, and until recently, this has been the traditional classification of Burseraceae [3]. Although recent molecular phylogenetic reconstructions are challenging the recognized infrafamiliar classification of the family [4,5], there is not yet a consensus from DNA data or additional evidence regarding synapomorphies from anatomy or morphology. Thus, current Burseraceae tribes are these: Protieae, with fruits dehiscent and pyrenes free or connivent but separable, the mesocarp and pericarp usually carnose or sometimes dry (some *Protium*); Boswellieae (=the autonym Bursereae), with fruits dehiscent consisting of pyrenes more or less connate but usually detachable from each other or separating when mature, the mesocarp and pericarp dry and dehiscent; and Canarieae, with fruits indehiscent and pyrenes more or less and often entirely connate, always forming a plurilocular putamen, pericarp carnose, rarely more or less dry.

Despite the importance of fruit structure in Burseraceae, there are few references in the literature regarding fruit anatomy. In general, literature on Burseraceae anatomy is scant [3]. In the tribe Canarieae, the most important of those references is Lam’s description of a three-winged bony axial “intrusion” in the fruit of *Canarium* [6], a characteristic Lam considered important enough to divide Canarieae into two subtribes based on the presence or absence of the intrusion. Another contribution to fruit structure in Canarieae was Cuatrecasas’s work on Neotropical *Dacryodes*, where he described “carpelar sutures” in the fruit [7] and an endocarp with “only one seminiferous cavity and one tiny, appressed sterile cell” [8]. The sterile cell has been used as part of the description of a New World species of *Dacryodes* as an articulated plate [9].

The present study documents fruit anatomy in tribe Canarieae, with the goal of a better understanding of what has been vaguely defined for a long time as a “degree of connation of the pyrenes”, which traditionally has formed the basis for Burseraceae infrafamilial classification. Understanding Burseraceae fruit anatomy will help elucidate the generic limits and phylogenetic relations in this tribe, as well as relations among Burseraceae tribes. To accomplish this, first it is important to review some definitions.

Burseraceae fruits have been considered as drupes [8,10], “more or less drupaceous” [11], or “variations on the drupe” [3]. According to Spjut [12], the Burseraceae have two fruit types, drupes and nuculania; in his system, a drupe is composed of a fleshy pericarp, with generally one to five stones (a stone is defined as a hard shell that encloses one or more seeds) whereas a nuculanium is a simple fruit with a dry pericarp and a hard endocarp, of one or more stones, the external layer(s) crustaceous or fibrous or coriaceous, indehiscent (*Garuga*) or dehiscent (*Boswellia*). To this last category belong the fruits of the subtribe Boswelliinae Daly, related to Canarieae in molecular studies [5,13]. These definitions correspond to those of Roth [14], in which drupes comprise a highly developed fruit type characterized as an indehiscent sarco-sclerocarpium, “where the pericarp reaches its classical subdivision into exo-, meso- and endocarp”. For this study, tissues were distinguished and classified based on the previous definitions of Spjut and Roth: “The exocarp is formed by the outer epidermis, but subepidermal layers may also be included; the mesocarp, generally composed of parenchyma, consists of large cells which may possibly be elongated radially. The endocarp is hard and consists either of sclereids or of fibers or of a mixture of both”.

## 2. Results

### 2.1. Canarium *L.*

Looking at the macroscopic characters of the endocarp, it is possible to see the presence of a thick bony layer separating the locules, corresponding to the axial intrusion described by Lam (Figure 1A,B). A cross-section of the endocarp reveals one developed locule and two reduced aborted locules. Each carpel has a layer surrounding it, probably corresponding to the inner epidermis.

The microsection (Figure 1C–G) shows the epidermis, ca. 4 µm thick, with a very dense hypodermis in 4–5 rows, composed of cells with resin-like deposits, densely packed towards the epidermis. Progressing inward is the mesocarp, composed by a broad layer of parenchyma, ca. 5 mm, with scattered cells with resin-like contents associated with secretory canals. Then, there is a thin layer of smaller parenchyma cells that abruptly changes to a lignified layer (endocarp), composed of sclereids with thick, symmetric, compact cell walls with some intracellular spaces.

### 2.2. Trattinnickia *Willd.*

The endocarp is bony and corrugate with two inconspicuous longitudinal sutures and the pyrenes are completely fused and not s separable by hand. The two samples studied here, *T. rhoifolia* Willd. and *T. burserifolia* Mart. (Figure 2A–D), have two and three locules, respectively. In this case, where three locules are present, both are equal in size, but only one develops a seed. No intercarpellar tissue (axial intrusion) was observed.

In the cross-section of the fruit (Figure 2E–I), *Trattinnickia* presents an epidermis ca. 6 µm thick, covered by a thick cuticle; no resinous hypodermis was observed. The mesocarp is characterized by two layers, the first composed of a dense layer of parenchyma cells ca. 30 µm thick, with resin and resin canals associated with vascular bundles, and the second inner layer has broader parenchyma cells without resin canals or vascular tissue, and possibly belongs to the endocarp, which is ca. 50 µm thick. This inner parenchymatic layer is slightly attached to the lignified layer of the endocarp, and is composed of symmetrical, compact lignified cells, very similar to those found in *Canarium*.

### 2.3. Dacryodes Vahl

An articulated plate was observed in Asian *D. rostrata* (Blume) H.J. Lam and Neotropical *D. edilsonii* Daly. The articulated plate is conspicuous once the pericarp is removed, and with little pressure, it opens, revealing another suture dividing it longitudinally, although it is more conspicuous from the internal side (Figure 3C). At the very apex of the fruit, the three sutures—two that compose the articulated plate—are slightly opened and seem to be supported by a fibrous tissue (Figure 3A,B).

The micro cross-section of the fruit reveals a lignified one-layer epidermis, radially elongated, ca. 30 µm thick, followed inward by a two-layered hypodermis with lignified cell walls, ca. 35 µm thick. A homogeneous mesocarp with vascular bundles follows, although the cells with a resin-like content are concentrated towards the epidermis, and further inward there is a region of non-lignified symmetrical cells of a different color and with a spongy appearance, ca. 215 µm thick, without vascular tissue, probably corresponding to the endocarp. Next follows the lignified layer of the endocarp, with thick lignified cells, ca. 570 µm thick.

### 2.4. Pseudodacryodes Pierlot

The dried herbarium sample preserved a cartilaginous consistency, converted to coriaceous-spongy when the fruit was rehydrated with heat. No sutures between the two carpels were found, and it was not possible to separate the exo-, meso-, and endocarp by hand. The tissue that surrounds the locules is the same as that which separates them, bearing two vascular bundles in the center. The endocarp is brown-red and does not have a lignified appearance (Figure 4A–D).

The microsection (Figure 4E–I) reveals a thin epidermis ca. 1 µm thick with a visible cuticle and without a hypodermal layer. The mesocarp, ca. 50 µm thick, has broad asymmetrical parenchyma, which has sparse vascular bundles ca. 2 µm diameter, surrounded by parenchyma cells with resin-like contents. Next, follows an endocarp unusual in Canarieae, formed by three layers: first, a layer of symmetrical parenchyma ca. 10 µm thick that contains high amounts of resin-like content, followed by a brilliant red-colored layer, apparently secreted from the previous layer, without cell walls or traces of lignification, and the innermost layer, ca. 2 µm thick, with the same appearance as the first layer of the endocarp but staining black, similar to the one found in *Santiria*. This innermost layer is also present in the zone that fuses the two carpels and surrounds the two vascular bundles that nurture the seeds, probably corresponding to the inner epidermis.

### 2.5. Santiria Blume

The two species studied, *S. laevigata* Blume and *S. trimera* (Oliver) Aubrév., show different degrees of compression of the carpels (Figure 5A–C).

The cross-section of the fruit (Figure 5E–H) shows an epicarp with a thin cuticle, and an epidermis ca. 25 µm with glands secreting a resin-like content, produced just below the epidermis. The mesocarp, ca. 1.5 mm thick, is also composed of broad parenchyma containing sparse secretory canals associated with vascular bundles. As in *Canarium*, towards the endocarp, the parenchyma become smaller, and the vascular tissue disappears. The hard layer of the endocarp is easily detachable from the adjacent small parenchyma cells, and it has two layers. The first is lignified and intricate, becoming compact towards the locule, ca. 150 µm thick. The innermost tissue is a thin single layer of black-staining tissue similar to the one found in *Pseudodacryodes*.

### 2.6. Rosselia Forman

The cross-section of the pyrene shows the sutures between the aborted locules, but the aborted locules are not fused and are detachable by hand (Figure 6A–C), forming two articulated plates with conspicuous apices. The microsection was performed on an immature fruit, thus what can be reported is that the fruit has a thin cuticle and a non-lignified epidermis on the epicarp, with few cells with resin-like contents present in the hypodermis, although no cells with resin-like contents were found in the mesocarp. The endocarp is thin but lignified, with cells that are very compressed, symmetrical, and with thick cell walls, similar to those of *Dacryodes*, although the endocarp seems to have two layers, the innermost (probably the inner epidermis) also lignified and thin (Figure 6D,E).

### 2.7. Haplolobus H. J. Lam

It is difficult to distinguish the three carpels, because the two aborted locules are very compressed and inconspicuous. A closer examination of the cross-sections reveals the presence of imbricate structures in the endocarp. Indeed, removing the thin exo- and mesocarp, three sutures are clearly visible. One of the aborted locule’s sutures overlaps the other two, and the other undeveloped locule goes from below the overlapping locule ending over the developed locule (Figure 7D–F). *Haplolobus* has a lignified epidermis ca. 1 µm thick, a subjacent two to three cell layer with resinous parenchyma and vascular bundles that is ca. 1–2 µm thick, followed within by a lignified layer ca. 4 µm thick with vascular bundles and resin cells toward the endocarp, corresponding to the highly reduced mesocarp. Next follows the endocarp with two layers: one layer of parenchyma, with no cells with resin-like contents, secretory canals, or vascular bundles, ca. 5 µm thick, and finally the innermost layer, lignified without resin-like contents, ca. 4 µm thick.

### 2.8. Scutinanthe Thwaites

The species studied here, *S. brunnea* Thwaites, indeed has a bony endocarp, and there is no evidence of articulation among the carpels, although with the compressed aborted locule, it resembles the pyrene of *Dacryodes* (Figure 8A). The microsection shows a pericarp without cuticle, with no evidence of cells with resin-like contents or secretory canals through the pericarp and mesocarp. Instead, there are intercellular spaces giving a ‘spongy’ appearance, and as described by Leenhouts, there are long palisade cells [10]. The endocarp is also different from the other genera studied: it is lignified and has at least two layers, the first with symmetric thick-walled cells with a resin-like content (staining black), and the second and innermost layer with cells forming a dense reticulate layer.

### 2.9. Garuga *Roxb.*

The fruit examined for this study has two developed and three aborted locules (Figure 9A). The epidermis has a thick cuticle ca. 15 µm thick and a very dense layer composed of the epidermis and subjacent resinous parenchymatic cells, compact towards the epidermis, ca. 200 µm thick. The mesocarp is rich in parenchyma with cells with resin-like contents and secretory canals associated with vascular bundles and lignified fibers. The endocarp’s histology varies between the undeveloped and the developed locules. The undeveloped locule, is surrounded by a lignified and not compacted layer, ca. 180 µm thick. The developed locules have a similar layer, about 1 mm thick, not compact (reticulate), with resin-like contents. Inside this layer, another lignified layer is present, ca. 180 µm thick and with a ‘palisade’ aspect that is elongated radially and with elongated cavities between the cells.

## 3. Discussion

The availability of material on this study permits the description of the tissues observed, their relative position, and particular characteristics, but not the tissue origin and/or the developmental biology of the fruit components.

Fruits of Paleotropical *Canarium* are large and ellipsoid, oblong or ovoid; the pericarp is carnose and very resinous [6], with an endocarp generally described as thick and bony [15] or woody [16]. The endocarp was described by Lam [6] as composed of pyrenes separated by a three-winged axial intrusion and covered by mesocarpal lids that open when the seeds are germinating. However, Ng reported that the endocarp of *Canarium* is fused to part of the stony inner layer of the mesocarp to form a thick-walled pyrene [17]. Similarly, Hill described the germination of *C. schweinfurthii* Engl., in which the ‘lid’ of a locule is present with a window-like fenestra that the embryo pushes and opens when germinating [18]. This last structure is more commonly named an operculum, and occurs in Burseraceae’s sister family, the Anacardiaceae, especially in the tribe Spondioideae [19].

*Trattinnickia* is a Neotropical group from humid forests of Central and South America, transferred from Protieae to Canarieae by Daly [20]. This genus is characterized by its hard bony endocarp and recent molecular studies reaffirm *Trattinninckia* as forming a strong supported clade with Neotropical *Dacryodes* [21].

*Dacryodes* is a genus that occurs in the Caribbean, Central and South America, Southeast Asia, and Africa. The presence of sutures along the carpels of *Dacryodes* has not been investigated anatomically but it has been mentioned in the literature as carpellar sutures or appressed sterile cells by Cuatrecasas [7,8] or as an articulated plate [9]. The later definitions refer to the structure composed of the carpels whose locules are aborted and become compressed and reduced.

The fruit of the rare monotypic genus from Central Tropical Africa *Pseudodacryodes leonardiana* has two locules, of which one develops a seed. The aborted locule, however, is not reduced as dramatically as *Dacryodes* and most of the other genera of Canarieae. The fruit of *Pseudodacryodes* is described as thin, not bony, with resinous pericarp and an eccentric intrusion separating the two locules [22].

As currently circumscribed, *Santiria* is a Paleotropical genus from the Western Malaysian region, the Philippines, the Moluccas, New Guinea, and Africa. *Santiria* fruits are similar in structure to those of *Dacryodes*: there are two sutures that evidence the presence of an articulated plate (which develops from the two undeveloped carpels), but the difference is that in *Santiria*, the articulated plate is oriented towards the base of the fruit, the result being that the stigma is distinctly excentric in the fruit [6].

*Rosselia bracteata* Forman is a recently described monotypic genus from New Guinea. *Rosselia* fruits have a coriaceous pericarp, with fibers running longitudinally, and the pyrene is similar to those of *Dacryodes* and *Santiria*, due to the presence of articulated plates; the ones in *Rosselia* are similar in shape and position to those of *Santiria*, but they are less fused to the developed locule, and the apices of the carpels remain free.

*Haplolobus* is a Paleotropical genus from Eastern Malaysia. The fruit of *Haplolobus* has been considered dry by some authors [23] or more or less dry [6] because of its membranaceous pericarp and really thin mecocarp. However, in *Haplolobus*, the articulated plate differs from that present in *Dacryodes*, *Santiria*, and *Rosselia* in that the former is not detachable from the fruit, and the two sutures are somewhat imbricate while in the latter genera, the articulated plate is detachable by hand and does not overlap.

*Scutinanthe* is a small genus with two species in Malaysia. It is the only genus in Canarieae with pentamerous flowers. The fruits have a fleshy pericarp, and the mesocarp has long radially arranged palisade cells; the pyrene is hard and bony, usually with two locules, one strongly reduced [10]. However, in *Haplolobus*, the articulated plate differs from that present in *Dacryodes*, *Santiria*, and *Rosselia* in that the former is not detachable from the fruit, and the two sutures are somewhat imbricate while in the latter genera, the articulated plate is detachable by hand and does not overlap.

*Garuga* is a small genus with four species in India and Southeast Asia. It currently remains in an unnamed subtribe in the Bursereae [24]. The fruit has one to five seeds, a carnose pericarp, and one to five small pyrenes that are gibbous, bony, thick, and resinous (Lam 1932a).

The endocarp is the most distinct part of the drupe [14] and it is indeed the most variable anatomical structure in Canarieae fruits. A study of Anacardiaceae fruits by Wannan and Quinn describes the Spondias-type, “which is composed of a mass of usually strongly lignified and irregularly oriented sclerenchyma” [19]. The authors also conclude that “the occurrence of valves or opercula that are dislodged at germination, appears to be restricted to this type of endocarp” but recognize this same fruit type in Canarium, suggesting this character is plesiomorphic.

Observations of the exterior of the pyrenes and cross-sections in this study show two different fruit morphologies (Table 1). The first group, composed of *Canarium* and *Trattinnickia*, has a very thick and bony endocarp, and the sterile locules are asymmetric, but the carpels are not compressed (that is, the abortion of locules does not affect the symmetry of the pyrene). *Canarium* differs from *Trattinnickia* by the presence of an intercarpellar tissue (Lam’s axial intrusion). Overall, there is a close congruence of fruit characters between *Trattinnickia* and *Canarium*, including the presence of very hard endocarps and a similar distribution of epicarp and mesocarp. Additionally, some species of *Canarium* and all *Trattinnickia* (but also *Garuga*) have a corrugate endocarp. The second fruit type is represented by *Santiria*, *Dacryodes*, *Rosselia*, and *Haplolobus*, sharing the presence of an articulated plate, compressed and displaced abortive carpels, and a smooth endocarp. *Scutinanthe* exhibits an intermediate fruit type, with fused bony pyrenes, no axial intrusion, and no evidence of a separable articulated plate, although the aborted locules are compressed and displaced. Finally, the placement of *Pseudodacryodes* is intriguing as well, being the only fruit with a cartilaginous endocarp. Germination of indehiscent fruits or with delayed indehiscence is explained by Roth: “many of the so-called indehiscent fruits open “passively” by the withering of the pericarp or by the pressure developed from the germinating embryo” [14].

The tissues of the fruits within the Canarieae tend to repeat certain patterns, with some modifications in the thickness of the different layers and their degree of lignification. The epidermis is conspicuously lignified in *Dacryodes* and *Haplolobus* but without evident lignification in the other genera. There is a very evident cuticle in *Pseudodacryodes*, *Rosselia*, *Santiria*, and *Trattinnickia*. The epidermis of all genera studied except *Scutinanthe* is accompanied by a resinous hypodermal layer, composed of smaller and densely distributed parenchyma cells with a resin-like content.

The mesocarp was defined in this study as the parenchymatic layer with vascular bundles, located between the epidermis and the endocarp. All taxa studied (except *Scutinanthe*) have parenchymatous cells with a resin-like content, and vascular bundles associated with secretory canals distributed through the mesocarp. In general, the mesocarp is the thickest layer in the fruit, the reduced mesocarp of *Haplolobus* being the only exception.

The distribution of endocarp tissues in Canarieae and *Garuga* is similar and consists of a parenchymatous layer outside a lignified layer (no evidence of lignification was found in *Pseudodacryodes*). This parenchymatous layer stained differentially in *Dacryodes*, and it was very rich in cells with a resin-like content in *Pseudodacryodes*. An extra innermost layer of thin parenchyma, staining black, was observed in *Pseudodacryodes* and *Santiria*, corresponding to what some authors interpreted as an inner epidermis [14]. Additionally, the descriptions presented here agreed with those made by Wannan and Quinn for the fruit of *Canarium oleosum* (Lam.) Engl. [19].

This study, based on herbarium specimens, revealed some promising characters for the understanding of fruit evolution in this group, but fresh material is needed to perform more complete analyses, including histological series with a broader representation of species for each genus, in order to document the developmental biology of the endocarp.

This study used fruit samples from herbarium specimens with available material for all eight genera of traditionally defined Canarieae (*Canarium*, *Dacryodes*, *Haplolobus*, *Pseudodacryodes*, *Rosselia*, *Santiria*, *Scutinanthe*, and *Trattinnickia*). *Garuga*, indehiscent but with free pyrenes, was originally placed in Protieae but was informally placed in Bursereae [15]; this has been supported by molecular studies [13,24]. For each genus, few mature fruits were available except for Neotropical *Dacryodes* and *Trattinnickia*.

## 4. Materials and Methods

Fruits were cut in cross-section with a carpenter’s saw (*Dacryodes*, *Garuga*, *Haplolobus*, *Pseudodacryodes*, *Rosselia*, *Scutinanthe*, and *Santiria*) or a bone saw (*Canarium* and *Trattinnickia*). Half of the fruits were boiled, and the pericarp removed to observe the presence of sutures or any other evidence of carpellar fusion. The rest of the fruits were softened in ethylenediamine at 37.5% and/or softening solution (10:3:90 10% Aerosol DT: glycerine: water). The treatment with ethylenediamine was selected due to its advantages for hard structures and plant tissues, especially when hard and soft tissues are mixed [25]. Differential treatments in the concentration of the solution and time of the treatment were performed based on the thickness and hardness of the endocarp. The time of treatment ranged from 2 days in ethylenediamine for *Pseudodacryodes leonardiana* R. Pierlot to 40 days followed by heating the samples to 75 °F for 30 min for *Canarium* spp.

After the tissues were softened, and the samples were embedded in Paraplast X-tra© for 12 h. The hard blocks were sectioned using a Leica RM2135 microtome (Heidelberger, Leica, Germany), and stained following Sharman’s procedure [26], an efficient method to differentiate plant tissues in which cell walls stain blue-black, nuclei stain yellow to orange, starch grains appear black, and lignified cell walls (i.e., sclerenchyma) stain brilliant red.

Endocarp samples were photographed with an Olympus Bo71 stereo microscope attached to a SPOT RT digital camera (Meyer Instruments, Houston, TX, USA). The prepared slides were observed and photographed under a Leitz Laborlux D Microscope attached to a SPOT 6 Insight firewire digital camera (Meyer Instruments, Houston, TX, USA).

## 5. Conclusions

This anatomical study of Canarieae fruits clarifies the general distribution of the tissues that correspond to the exo-, meso-, and endocarp. The detailed arrangement and measurements of the tissues are documented here for the first time in almost all genera of the tribe. The evidence suggests that in all cases (except *Pseudodacryodes*), the endocarp has at least one layer of parenchyma cells followed inward by a lignified layer, and in some cases, an inner epidermis is present. All Canarieae exhibit resinous fruits and evidence of carpellar sutures, and in *Dacryodes*, *Santiria*, *Haplolobus*, and *Rosselia*, an articulated plate is present. *Garuga* exhibits the same tissue arrangement in each pyrene, but the pyrenes are free in the (endocarpal) parenchyma.

The characters described may serve as basic information to support the generic limits on Canarieae.

## Figures and Tables

**Figure 1 plants-11-00253-f001:**
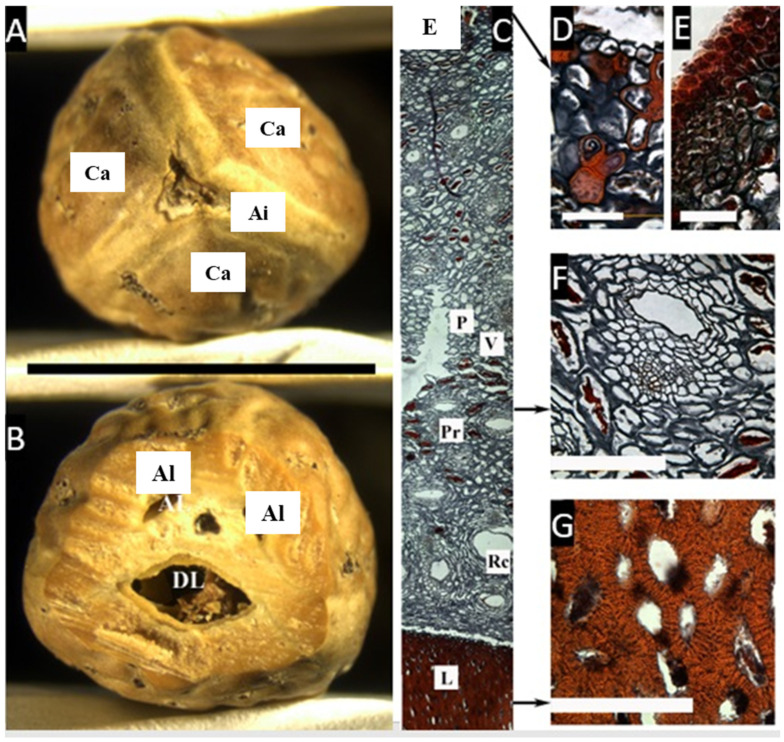
(**A**,**B**) Pyrene of *Canarium asperum*. (**A**) Upper view, showing three carpels (Ca) and the axial intrusion (Ai). (**B**) Cross-section with one developed locule (Dl), and two aborted locules (Al). (bar = 2.5 cm). (**C**–**G**) Cross microsection of the pericarp (**C**) Complete cross-section showing the epidermis (E), parenchyma (P), vascular tissue (V), parenchymatic cells with resin-like contents (Pr), secretory canals (Rc), and lignified cells in the pyrene (L). (**D**,**E**) Detail of the epicarp showing the epidermis and a dense hypodermis. (**F**) Detail of the mesocarp with cells with a resin-like content and a secretory canal. (**G**) Heavy lignified cell walls in the endocarp. ((**A**,**B**), bar = 2.5 cm; (**C**), bar = 50 µm; (**D**–**G**), bar = 100 µm; (**A**,**B**) from Mabulid 6860, NY; (**C**–**G**) from Merril 5857, NY).

**Figure 2 plants-11-00253-f002:**
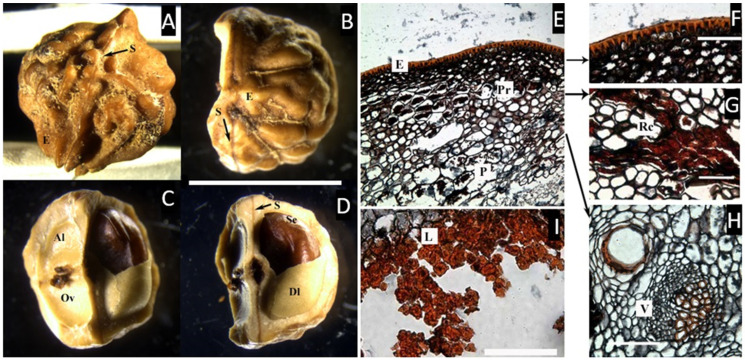
(**A**) Pyrene of *Trattinnickia rhoifolia* in lateral view, with one inconspicuous longitudinal, suture (S). (**B**–**D**) *T. burserifolia*. (**B**) Apical view of the bony endocarp (E) with two sutures (S) and remains of the fibrous mesocarp. (**C**,**D**) Longitudinal views of the endocarp showing two aborted locules (Al), each with one ovule (Ov), and one developed locule (Dl) bearing the seed (Se). (bar = 1 cm). (**E**–**H**) Cross microsections of *Trattinnickia rhoifolia*. (**E**) Cross-section of the outer layers of the pericarp showing the epidermis with a cuticle (E) and parenchymatous mesocarp (P) with some cells with resin-like contents (Pr). (**F**) Detail of the epicarp. (**G**) Detail of the mesocarp with resin canals (Rc). (**H**) Detail of a vascular bundle (V). (**I**) Endocarp with a parenchyma layer lacking resin-like contents, and an internal layer of symmetric, compact, lignified cells (L). ((**A**–**D**), bar = 1 cm; (**F**–**I**), bar = 50 µm. (**A**,**E**–**I**), from Cremer 2893, NY; (**B**–**D**), from Foldats 43505, NY).

**Figure 3 plants-11-00253-f003:**
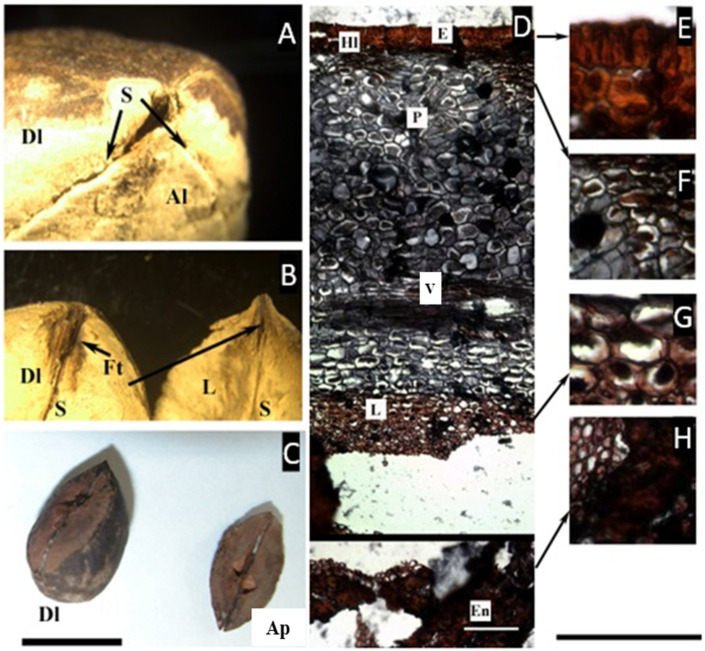
(**A**–**C**) Pyrene of *Dacryodes edilsonii*. (**A**) Apical view showing the sutures (S), the developed locule (Dl), and the articulated plate, composed of the two aborted locules (Al). (**B**) When separated from each other, the articulated plate and the developed locule are separated by a longitudinal suture and fibrous tissue (Ft). (**C**) Pyrene of *D. rostrata* illustrating the relative size of the two aborted carpels that form the articulated plate (Ap), compared with the developed locule (Dl). (**D**–**H**) Microsection of *Dacryodes edilsonii*. (**D**) Complete cross-section of the pericarp showing the epidermis (E), hypodermal-lignified layer (Hl), parenchyma (P), vascular bundle (V), and parenchymatic layer of the endocarp (P). (**E**) Detail of the epicarp. (**F**) Detail of the mesocarp showing cells with a resin-like content. (**G**) Detail of the parenchymatic layer of the endocarp. (**H**) Lignified compact layer of the endocarp facing the locule (En). ((**C**), bar = 1.5 cm; (**D**–**H**), bar = 50 µm. (**A**–**H**) from D. C. Daly 11917, NY).

**Figure 4 plants-11-00253-f004:**
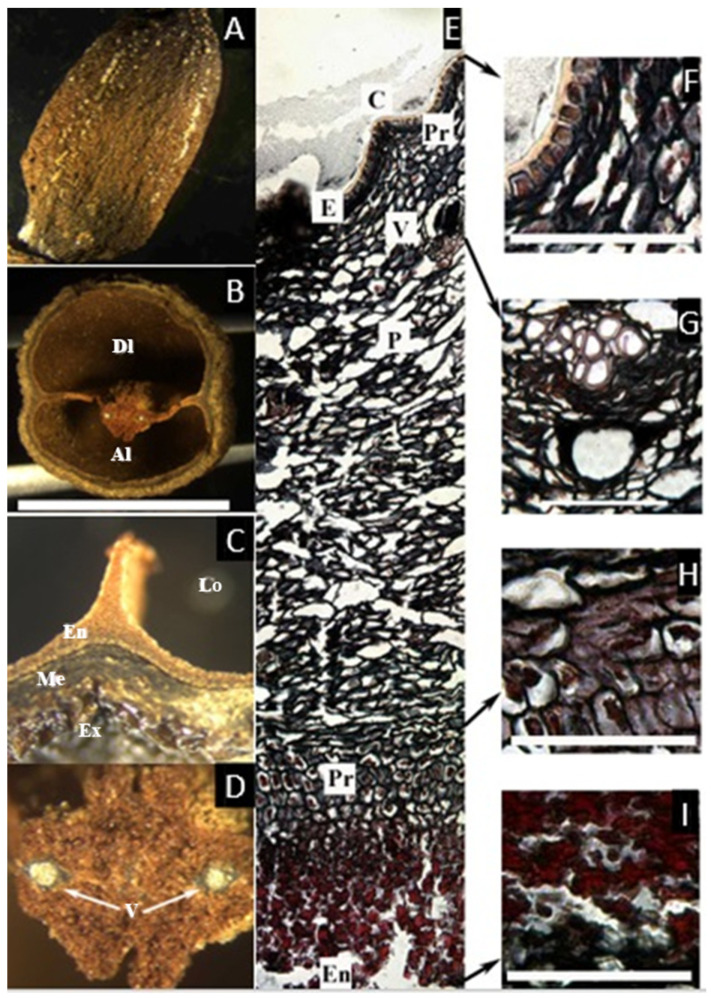
(**A**–**D**) Fruit of *Pseudodacryodes leonardiana*. (**A**) Fruit in lateral view. (**B**) Cross-section of fruit showing the developed (Dl) and aborted (Al) locules of similar size. (**C**) Closer view of the exocarp (Ex), mesocarp (Me), endocarp (En), and locule (Lo) (**D**) Two vascular bundles (V) in the center of the fruit. (**E**–**I**) Microsection of *Pseudodacryodes leonardiana*. (**E**) Complete cross-section of the pericarp showing the translucent cuticle (C), the epidermis (E), apparently containing resin, parenchyma with resin-like content (Pr), vascular bundles (V), parenchyma (P), and the endocarp (En). (**F**) Detail of the epicarp. (**G**) Detail of the resin canal associated with a vascular bundle. (**H**) Detail of the resin-like producing layer of parenchyma. (**I**) Detail of the innermost layer, staining black. ((**A**–**D**), bar = 8 mm; (**E**–**I**), bar = 50 µm; (**A**–**D**) from Pierlot 2879, BR).

**Figure 5 plants-11-00253-f005:**
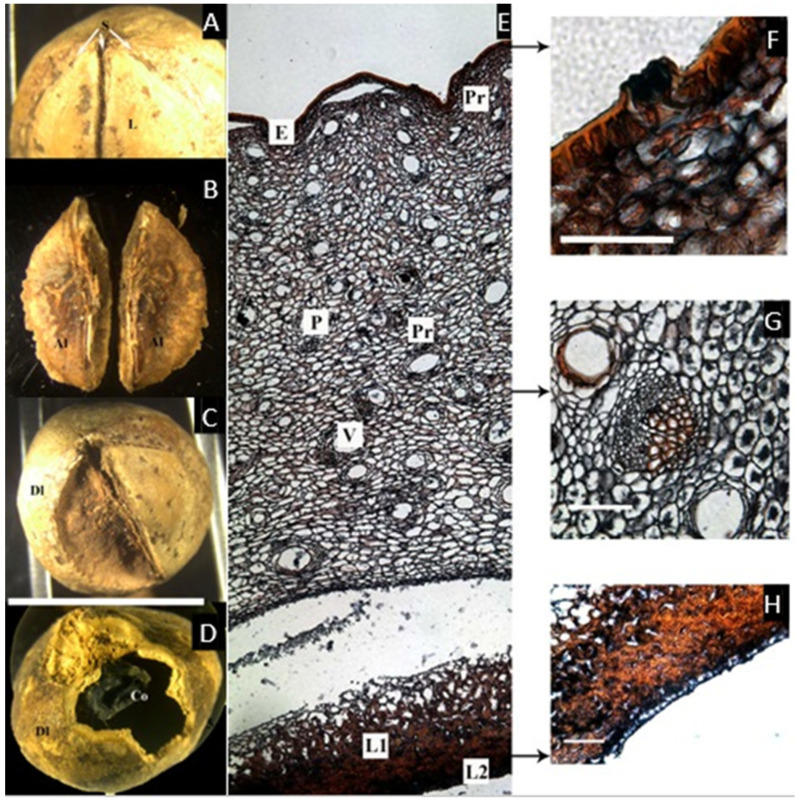
(**A**,**B**) Pyrenes of *Santiria laevigata*. (**A**) Apical view showing sutures (S) and the lid (L) or articulated plate. (**B**) Lateral view of the articulated plate showing two aborted locules (Al), easily detached from each other along the longitudinal suture. (**C**,**D**) Pyrenes of *S. trimera*. (**C**)**.** Apical view of the developed locule (Dl) with the articulated plate removed. (**D**) Developed locule (Dl) opened to show lobed and folded cotyledon (Co). (**E**–**H**) Microsection of *S. laevigata.* (**E**) Complete cross-section of the pericarp showing epidermis (E), parenchyma with resin-like content (Pr) that secretes into glands; mesocarp with sparse vascular bundles (V) associated with secretory canals (Pr), and endocarp that has a compact lignified layer (L1) and black-staining inner epidermis (L2). (**F**) Detail of the epidermis and a gland. (**G**) Detail of a vascular bundle in the mesocarp and associated secretory canal. (**H**) Detail of the lignified endocarp and inner epidermis ((**C**), bar = 1 cm; (**F**–**H**) = 50 µm; (**A**,**B**,**E**–**H**) from E. J. H. Corner 26046, NY; (**C**,**D**) from McPherson 15445, NY).

**Figure 6 plants-11-00253-f006:**
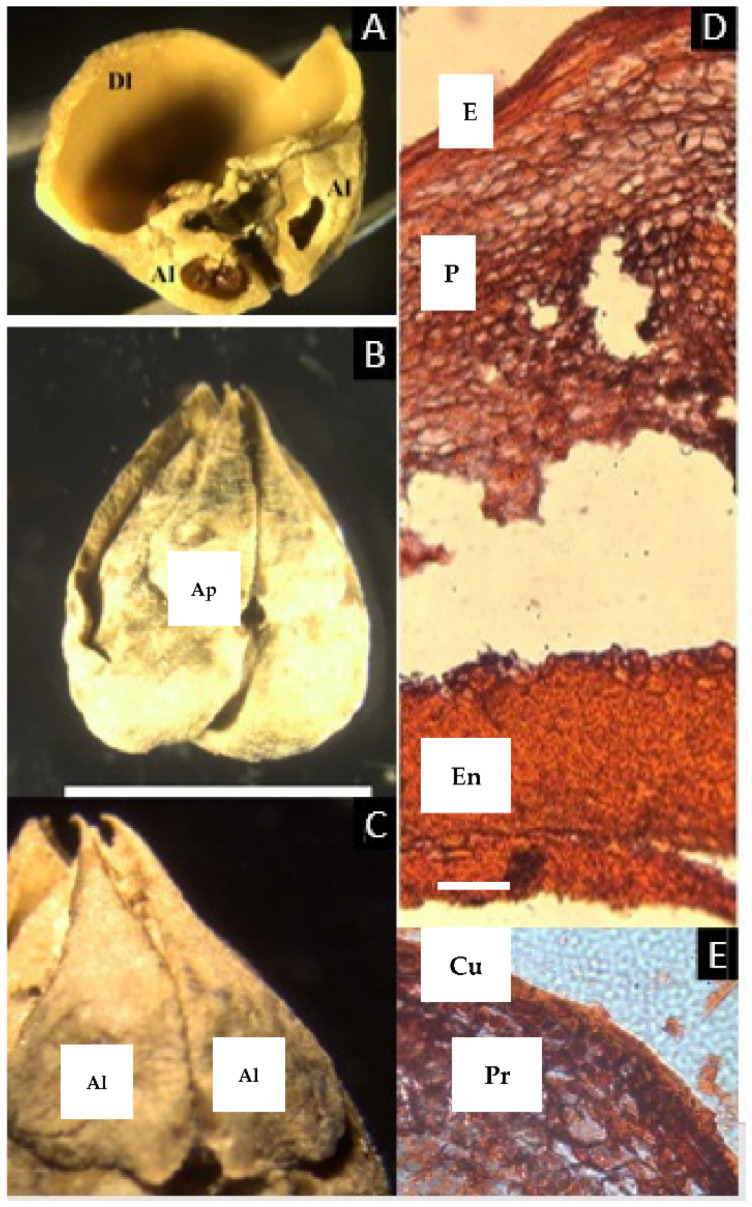
(**A**–**C**) Pyrene of *Rosselia bracteata*. (**A**) Cross section of the pyrene showing the developed locule (Dl) and two aborted locules (Al). (**B**) Lateral view of the pyrene showing the articulated plates (Ap). (**C**) Detail of the free apices of the carpel that form the articulated plate (Al). (**D**,**E**) Microsection of immature fruit. (**D**) Complete cross-section of the pericarp showing epidermis and mesocarp with parenchymatic cells (P) and an endocarp with two lignified layers (En). (**E**) Detail of the epicarp showing the cuticle (Cu) and cells with resin-like content in the hypodermis (Pr). ((**B**), bar = 5 mm; (**D**), bar = 50 µm; (**A**–**E**) from *Brass 24545*, L).

**Figure 7 plants-11-00253-f007:**
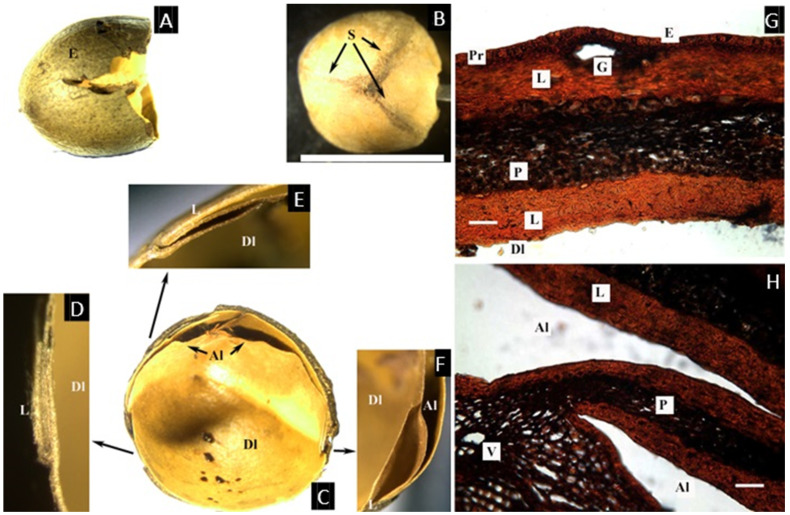
(**A**–**F**) Fruit of *Haplolobus floribundus*. (**A**) Fruit showing the fragile membranaceous consistency. (**B**) Apical view of the pyrene showing three carpels divided by three sutures (S). (**C**) Cross-section of the fruit showing the developed locule (Dl), aborted locule (Al). (**D**–**F**) Detail of the area joining the carpels. (**D**) An aborted used. (**E**) One aborted carpel overlapping the other. (**F**) An aborted locule overlapping the developed locule. (**G**,**H**) Microsection of *Haplolobus floribundus*. (**G**) Complete pericarp cross-section showing lignified epidermis (E), hypodermis with a thin layer of parenchyma cells with resin-like content (Pr), lignified mesocarp (L), and the endocarp composed of an exterior layer of parenchyma (P) and interior compact lignified layer (L) facing the developed locule (Dl). (**H**) Detail of tissues surrounding the two aborted locules (Al) and the central vascular bundle (V). Notice the parenchyma layer (P) between the lignified endocarp (L). ((**B**), bar = 1 cm; (**G**), bar = 50 µm; (**H**), bar = 5 µm; (**A**–**E**) from *L. J.*
*Brass 29262*, NY).

**Figure 8 plants-11-00253-f008:**
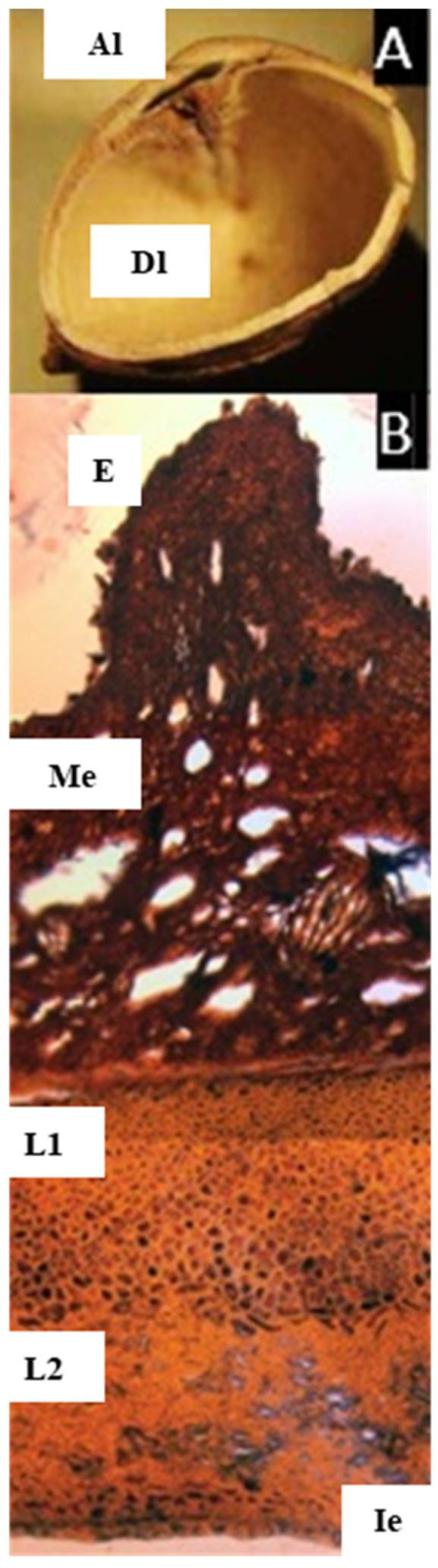
(**A**) Pyrene of *Scutinanthe brunnea*. Cross-section showing one developed locule (Dl), and the longitudinally compressed aborted locule (Al), lacking articulation. (**B**) Microsection of the pericarp of *Scutinanthe brunnea*. Showing the epidermis (E), parenchymatic mesocarp with intercellular spaces (Me), a two-layered lignified endocarp (L1, L2), and the inner epidermis (Ie). ((**A**,**B**) from s.n. 1149, NY). ((**A**,**B**), bar = 200 µm).

**Figure 9 plants-11-00253-f009:**
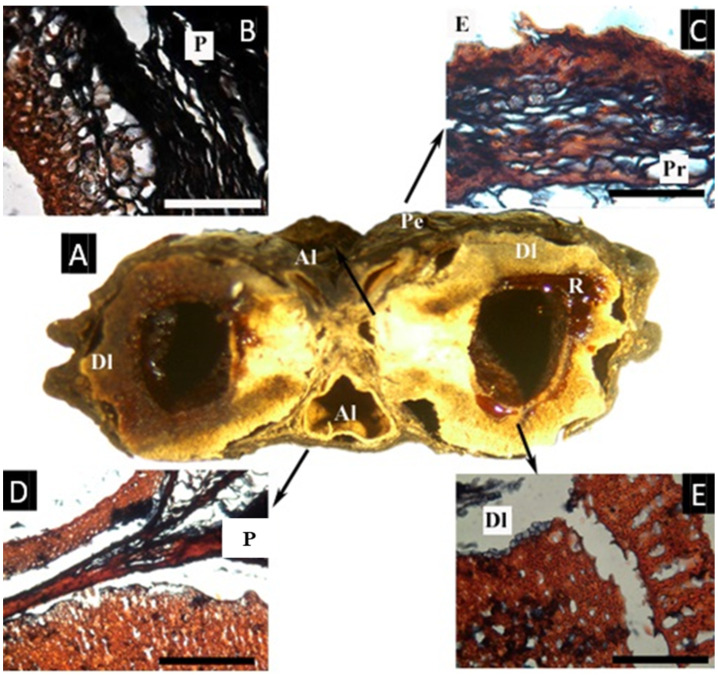
(**A**–**E**) Fruit and microsection of *Garuga floribunda*. (**A**) Cross-section of the fruit, showing the pericarp (Pe), aborted locules (Al), and developend locules (Dl) with resin-like secretion (R). (**B**) Detail of one aborted locule with a lignified innermost layer, surrounded by parenchyma of endocarpal origin, lacking vascular tissue. (**C**) Detail of the epidermis with a thick cuticle, and a parenchymatous hypodermis with cells with a resin-like content. (Pr). (**D**) Parenchymatous layer (P) between the two aborted locules. (**E**) Detail of the endocarp surrounding one developed locule (Dl), with two layers of lignified cells, the outermost similar to that present in the undeveloped locules, the innermost layer radially elongated. ((**A**), bar = 5 mm; (**B**–**E**), bar = 5 µm; (**A**–**E**) from D. D. Soejarto *7742*, NY).

**Table 1 plants-11-00253-t001:** Comparison of fruit characters for the taxa sampled (+ = present, - = absent).

Taxa	Axial Intrusion	Lignified Hypodermis	Articulated Plate	Pyrene Surface	Outermost Layer in Endocarp	Innermost Layer in Endocarp	Inner Epidermis in Developed Locule
*Canarium* *asperum*	+	-	-	smooth/ rugose	lignified	lignified	not observed
*Dacryodes* *edilsonii*	-	+	+	smooth	parenchymatic	lignified	not observed
*Garuga* *floribunda*	-	-	-	rugose	lignified	lignified	not observed
*Haplolobus* *floribundus*	-	-	-	smooth	parenchymatic	lignified	not observed
*Pseudodacryodes leonardiana*	-	-	-	smooth	parenchymatic	parenchymatic	observed
*Rosselia* *bracteata*	-	-	+	smooth	lignified	lignified	not observed
*Santiria* *laevigata*	-	-	+	smooth	-	lignified	observed
*Scutinanthe* *brunnea*	-	-	-	smooth	lignified	lignified	observed
*Trattinnickia rhoifolia*	-	-	-	rugose	parenchymatic	lignified	not observed

## Data Availability

Not Applicable.

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
