# Peer review of "Fruit Anatomy of the Canarieae (Burseraceae)"

_plants, 2022, doi:10.3390/plants11030253_

Round 1

Reviewer 1 Report

The manuscript brings interesting information on fruit anatomy in tribe Canarieae. Overall, the manuscript is well-written. The anatomical figures are of very good quality. The results are clearly presented. I would suggest to separete results from discussion. The approach to methodology used is appropriate however some information in methods is missing (Lines 392 and 395; country of production should be added).

Specific comments follow:

Figure Legend 1, Line 88; (E) is used for epidermis not (Ep). Remove bar from line 87, bars are described in line 91. 

Bars should be added to Figures 6D and 8.

References should be formatted in accordance with the journal requirements. 

Author Response

Dear reviewer,

Many thanks for your comments and suggestions. I recently updated the revised version, including the separation of results and discussion, and adding the coutry of production of the instruments used.

Best wishes, 

Reviewer 2 Report

1. 3.1 subtitle could be deleted

2. The fruit traits examined in this study only showed the anatomy, it  will be impressive if more information provided, such as fruit size, color, and others.

3. The introduction section could be shorten if possible.

Author Response

Dear reviewer,

Many thanks for your comments and suggestions. I recently updated the revised version, including the removal of the subtitle in the results section.

Best wishes, 

Reviewer 3 Report

The article deals with taxonomic characters of Burseraceae, with a focus on a drupe pyrenes structure as a good descriptive character. The author chose several species from herbarium collections and did a morpho-anatomical examination. The article is well written and for the people of this field can be interesting.

The limitations of this work are mentioned by the author herself- there are no biological repeats, and only 9 species from a large family were examined. However, the work can indeed serve as a good basis for further studies. Also, due to the technical difficulty of this work, the quality of the images is quite impressive. I think the study should be published.

Several small remarks:

L 22- “Infrafamiliar” should be “intrafamiliar”

L23- for the sake of general reading audience, please explain what’s “pyrena/pyrene"

In Table 1 please write the full latin names+ genus examined for more clarity

It could be nice to add a scheme describing the different structures, but it's only a suggestion. 

Author Response

Dear reviewer,

Many thanks for your comments and suggestions. I recently updated the revised version, including the definition of pyrene and the full names of taxa in Table 1.

Regarding the word "infrafamiliar", I respectfully want to mention that it's more specific, since it refers to the hierarchy within the relationships of the tribes and in the family.

Best wishes,